# Preclinical Implementation of matRadiomics: A Case Study for Early Malformation Prediction in Zebrafish Model

**DOI:** 10.3390/jimaging10110290

**Published:** 2024-11-14

**Authors:** Fabiano Bini, Elisa Missori, Gaia Pucci, Giovanni Pasini, Franco Marinozzi, Giusi Irma Forte, Giorgio Russo, Alessandro Stefano

**Affiliations:** 1Department of Mechanical and Aerospace Engineering, Sapienza University of Rome, Eudossiana 18, 00184 Rome, Italy; fabiano.bini@uniroma1.it (F.B.); missori.1894850@studenti.uniroma1.it (E.M.); franco.marinozzi@uniroma1.it (F.M.); 2Institute of Bioimaging and Complex Biological Systems—National Research Council (IBSBC—CNR), Contrada Pietrapollastra-Pisciotto, 90015 Cefalù, Italy; gaiapucci@cnr.it (G.P.); giusiirma.forte@cnr.it (G.I.F.); giorgio-russo@cnr.it (G.R.); alessandro.stefano@cnr.it (A.S.)

**Keywords:** radiomics, preclinical image analysis, image resolution analysis, early prediction

## Abstract

Radiomics provides a structured approach to support clinical decision-making through key steps; however, users often face difficulties when switching between various software platforms to complete the workflow. To streamline this process, matRadiomics integrates the entire radiomics workflow within a single platform. This study extends matRadiomics to *preclinical* settings and validates it through a case study focused on *early* malformation differentiation in a zebrafish model. The proposed plugin incorporates Pyradiomics and streamlines feature extraction, selection, and classification using machine learning models (linear discriminant analysis—LDA; k-nearest neighbors—KNNs; and support vector machines—SVMs) with k-fold cross-validation for model validation. Classifier performances are evaluated using area under the ROC curve (AUC) and accuracy. The case study indicated the criticality of the long time required to extract features from preclinical images, generally of higher resolution than clinical images. To address this, a feature analysis was conducted to *optimize* settings, reducing extraction time while maintaining similarity to the original features. As a result, SVM exhibited the best performance for early malformation differentiation in zebrafish (AUC = 0.723; accuracy of 0.72). This case study underscores the plugin’s versatility and effectiveness in early biological outcome prediction, emphasizing its applicability across biomedical research fields.

## 1. Introduction

Radiomics has become a valuable tool in medical imaging, enabling the extraction of quantitative features that support clinical decision-making, especially in oncology [1,2,3,4]. It provides insights into tumor heterogeneity, early treatment response [5], and patient prognosis [6,7,8,9]. The typical radiomics workflow includes key steps: target identification and segmentation, feature extraction and selection, and predictive model development using machine learning (ML) [10,11,12,13]. Despite the availability of various software for these tasks, transitioning between tools remains challenging [14,15,16,17]. To streamline this, matRadiomics was developed to integrate all workflow steps into a single environment [18]. Initially developed for clinical applications, matRadiomics has proven effective in simplifying radiomics analyses, particularly in positron emission tomography (PET) imaging for prostate cancer, by reducing dependency on multiple software tools [19]. Enhanced by Artificial Intelligence, it further improves decision-making and diagnostic accuracy in clinical settings [20,21].

As preclinical research gains prominence, there is an increasing need to adapt radiomics tools for these studies [22]. Preclinical imaging plays a critical role in understanding disease mechanisms and evaluating treatment efficacy [23]. However, these images often have a higher resolution than clinical images. Unlike tomographic techniques, such as computed tomography (CT), magnetic resonance imaging (MRI), and PET, which are used in both clinical and preclinical settings for large animals, such as mice, optical imaging is applied in zebrafish or cellular models to capture fine details [24,25]. These types of images, not being obtained by medical instrumentation, have a higher resolution than clinical images. For this reason, they present unique challenges in terms of data processing and feature extraction [26]. Feature extraction on large masks significantly increases computational time due to the high data volume. This study analyzed optimal radiomics settings to address the extended processing times while ensuring reliable outputs. To the best of our knowledge, no guidelines exist in this area. For this reason, our goal was to propose preliminary recommendations to optimize feature extraction from high-resolution preclinical images, balancing accuracy and computational efficiency. These preliminary guidelines will help researchers produce reliable data without prohibitive processing times, making radiomics more accessible for exploratory preclinical studies.

With this goal, we have extended matRadiomics to distinguish it from other radiomics tools by integrating several specialized features, designed specifically for preclinical radiomics workflows. First, it supports a wide range of image formats (JPG, TIFF, and PNG) not common in other platforms, allowing users to work with high-resolution images essential in preclinical studies, such as those of zebrafish models, which we used in our case study described below. In this way, unlike existing tools, like LIFEx and FeAture Explorer (FAE) [16,27], which focus on clinical data, matRadiomics extends beyond to handle cell-based and small-animal models, accommodating the specific imaging needs of these contexts. Additionally, matRadiomics incorporates Pyradiomics for standardized and reproducible feature extraction, adhering to Image Biomarker Standardization Initiative (IBSI) guidelines, thus reducing variability and enhancing comparability across studies [28,29]. Thanks to the integration of Pyradiomics and incorporating ML algorithms for the implementation of predictive models, it allows you to complete a radiomics study with the same platform without switching between software, unlike all other State-of-the-Art radiomics software. Furthermore, matRadiomics offers a streamlined workflow with advanced segmentation and feature extraction options specifically optimized for high-resolution preclinical images, which tend to require lengthy processing. By incorporating intensity normalization and voxel resampling, it addresses the computational challenges of large preclinical datasets. These improvements ensure that researchers can generate meaningful insights more rapidly, enabling faster turnaround times for exploratory studies and supporting translational research efforts. In particular, the tool’s support for simultaneous multi-region analysis further helps researchers conduct more complex analyses, which is crucial for biological studies where there are multiple targets to analyze (e.g., toxicity or radiopharmaceutical accumulated in various organs [22]). Through these advancements, matRadiomics significantly contributes to the field, enhancing usability, flexibility, and reliability in preclinical radiomics workflows. This makes it a valuable resource not only for image analysis but also for fostering collaborative research efforts: to expand the sample set, the software includes an “Import Study” function, which allows users to import the various parameter settings used to complete the radiomics study, enabling more immediate collaboration between different research institutions.

As anticipated, the feasibility evaluation of this extended version of matRadiomics is based on a case study involving zebrafish models. The zebrafish is a validated vertebrate model for disease, drug screening, target identification, and pharmacology [30]. It is suitable for studying a variety of different situations, such as genetics, cell biology, toxicology, and embryology, filling a scientific niche between in vitro models and higher organisms [31]. Concerning toxicity assays, Sipes et al. [32] reviewed the concordance of 55–100% between zebrafish and mammalian models in evaluating chemicals with toxic effects. So, the percent concordance across the mammalian species suggests that the response of zebrafish is on par with mammalian toxicity models and supports the utility of this model in toxicology research, as to identify drug targets or optimize clinically relevant treatments, such as radiotherapy [33]. Their widespread and increasing use in toxicological studies is mostly due to their genetic similarity to humans, with 70% of homology and 82% of orthologous human disease-related genes [34], their transparency, rapid reproductive cycle and embryonic development, high permeability to substances, low maintenance costs, and exemption from ethical protocols for zebrafish up to 120 hpf. These factors enable precise and real-time observations of biological malformations and organ toxicity, providing useful information in toxicity assessments [30,31,32,33]. Furthermore, zebrafish show high fertility: each pair can produce from around 200 to 300 eggs per week, thus permitting large-scale experimental analysis and, consequently, a higher statistical validity of the result [35].

Considering that the onset of malformations represents a visible response to a treatment, in this study, we assess the platform’s ability to discriminate “early” between different levels of biological malformations following radiotherapy treatment at 30 Gy. The analysis was performed at only 72 h post-fertilization (hpf), unlike other State-of-the-Art works analyzing images at 96 and 120 hpf [36,37,38,39], which are also the time points at which larvae show greater mobility and activity [40]. Therefore, our idea is to enable more timely predictions of treatment toxicity, without necessarily reaching later time points. In our analyses, high and low levels of malformation were detected at only 72 hpf. This early detection would also improve the efficiency of toxicity screening in drug discovery and development.

In conclusion, this study aims to establish matRadiomics as a reference tool not only for clinical applications but also for the broader field of biomedical research, facilitating the integration of radiomics into preclinical studies and expanding its potential to address key questions in disease modeling and drug development.

## 2. Material and Methods

The extended version of matRadiomics was enhanced to include the following functionalities: (i) create or import a preclinical radiomics study; (ii) import and visualize images in JPG, TIFF, and PNG formats; (iii) perform target segmentation using multiple methods, including importing pre-existing masks; (iv) extract features from multiple masks using Pyradiomics; (v) conduct feature selection; and (vi) Apply ML algorithms for predictive modeling.

### 2.1. Improvements in matRadiomics

The architecture of the tool remains the same as in the previous version. However, to make the radiomics analysis applicable in the preclinical context, new functions were introduced to compensate for the absence of certain image information typically available in clinical images (e.g., Digital Imaging and Communications in Medicine, DICOM, format). In particular, when information regarding voxel spacing and slice thickness is unavailable, unit values of [1,1,1] were assigned. This parameter is crucial for the correct execution of the feature extraction process using the Pyradiomics module, and for an accurate interpretation of the extracted features [41].

Upon initiating a radiomics analysis—either as a new study or by importing an existing one—it is now possible to import images in JPG, TIFF, and PNG formats.

In addition to the previously available manual and semi-automatic segmentation tools, users can now import externally created masks in JPG format. In cases of multiple segmentation, such as in fish where various body parts need to be segmented, each mask can be loaded sequentially with a unique name, facilitating easy identification in subsequent radiomics stages. When analyzing the first image study, the user must manually enter the name for each mask. After the initial feature extraction, the program stores these mask names, allowing them to be easily associated with masks from subsequent image studies by selecting from a list. It is also possible to derive new masks through subtraction operation, for example, by removing a specific organ, such as the heart, from the whole animal. Feature extraction is performed using the Pyradiomics module, which resulted in the extraction of 107 features, including shape features that describe the geometric properties of the regions of interest, which are critical to understanding anatomical variations; first-order statistics that provide information about the distribution of pixel intensities, reflecting the overall brightness and contrast of the image; and texture features that capture patterns and relationships in pixel intensities, providing valuable insights into tissue heterogeneity. Unlike the previous version, the new version allows for the extraction of features from multiple masks in series. The extracted features are displayed in the “Features Extraction” table, with each mask recognizable thanks to the associated name. Both the extracted features (in “.xlsx” format) and the masks (in “.mat” format) are saved within the study folder, organized by sample.

For the feature selection and prediction model selection, users can choose from several methods, including ReliefF [42], Point Biserial Correlation (PBC) [43], *t*-test, and LASSO [44,45]. In terms of prediction modeling, the software offers three classifiers: linear discriminant analysis (LDA) [46], k-nearest neighbors (KNNs) [47], and support vector machines (SVMs) [48]. Two methods are available for the model validation: k-fold cross-validation and stratified k-fold cross-validation.

Finally, to expand the number of samples, the software provides an “Import Study” section. This feature allows users to *collaborate* by importing a radiomics study that has already completed the feature extraction phase. The program automatically recognizes and stores the names of the analyzed masks upon import, making it easy to use when creating masks for new images to analyze.

To further clarify the improvements made in this release of matRadiomics, we highlight that the newly introduced features directly address the challenges faced in preclinical imaging. By allowing the import of various image formats and enabling the integration of external masks, the tool improves usability and flexibility. Furthermore, the ability to manage multiple masks during analysis simplifies the workflow, especially in complex preclinical studies where the analysis of multiple regions of interest is crucial. These advances not only improve the efficiency of radiomics analysis but also aim to facilitate collaboration between researchers through the functionality of importing external studies.

Figure 1 shows the workflow of matRadiomics, showing its basic characteristics.

### 2.2. Resolution Analysis for Radiomics Feature Extraction

During this study, it was observed that feature extraction for large masks resulted in a significant increase in computational time, likely due to the high resolution of the preclinical images. The literature provides limited guidance on this problem, other than recommendations to preprocess images to reduce dimensionality or use GPU to complete the task faster [26,49]. To address this, we conducted an analysis to identify the optimal settings for the radiomics study, ensuring faster feature extraction without compromising the integrity of the features.

The first step was to evaluate the time required for feature extraction. Specifically, we applied the radiomics workflow to images at their original size (in our case, 2560 x 1920) and a bin count of 64. Bin count refers to the number of bins used to group pixel intensity values. A value of 64 is commonly used for gray-level discretization to balance sensitivity and computational feasibility. Conversely, setting a bin count of 1 means using a single bin, making it easier to analyze gray-level related features. This value preserves the original gray-level values, allowing for a comparison that reveals how much influence binning has on feature stability and error rate. This approach allowed us to specifically examine the influence of gray-level features on feature variability, allowing for a more targeted analysis. We determined that, for a single mask, comparable in size to the image, approximately 15 min was required for feature extraction (specifications of the computer used for the extraction: processor, 11th Gen Intel(R) Core(TM) i7-1165G7 @ 2.80 GHz; RAM, 32.0 GB; 64-bit operating system, x64-based processor; graphics card, Intel(R) Iris(R) Xe Graphics, 4 GB RAM; graphics max dynamic clock, 1500 MHz; SSD, 1 TB). Given the huge number of masks and images in our case study, this time request was unfeasible, necessitating interventions to improve extraction speed.

To address the issue, small-sized masks of six zebrafish images were initially considered. A total of 107 features were analyzed. For all cases, we evaluated the magnitude of the relative percentage error for each feature and mask as follows:$$\delta_{i,j}=\left|\frac{v_{A\left(i,j\right)}-v_{E\left(i,j\right)}}{v_{E\left(i,j\right)}}\;\right|\cdot 100;\;$$
$$v_{A\left(i,j\right)}:observed\;value;\;v_{E\left(i,j\right)}:original\;value;\;i:features,\;j:\;masks$$

The mean of the relative percentage error among the masks for each type of feature was determined as follows:$$\delta_{i}=\frac{\sum_{j=1}^{m}\delta_{j}}{m};\;m=masks\;number\;for\;each\;features$$

And, finally, we calculated the average relative percentage error across all features:$$M=\frac{\sum_{i=1}^{n}\delta_{i}}{n};\;\;n=extracted\;features\;number$$

After this, we evaluated two different preprocessing methods: loading resized images and masks or applying the matRadiomics resampling tool after loading the images and the masks in their original size. The resizing function was implemented directly in the program’s code, adding a function to apply resizing to the images and masks upon import. Regarding resampling, no modifications were made to the existing code; instead, the preprocessing tool already provided by matRadiomics was utilized. The resampling function is provided by Pyradiomics, which applies the type of interpolation specified by the user for the image—in this case, stick-nearest neighbor—while always using nearest-neighbor interpolation for the mask. A reduction factor of 2.5 was applied in both cases. The choice of this factor represents a compromise between reducing the dimensionality of the images and minimizing information loss. A reduction by 1/2 did not significantly decrease the dimensionality, while a 1/3 reduction resulted in excessive loss of detail. Therefore, a reduction factor of 1/2.5 was selected to achieve a meaningful decrease in dimensionality without sacrificing too much information. During resizing, the voxel spacing was adjusted accordingly, directly in the program’s code, while for resampling, the aforementioned techniques were used, with no modifications made to the existing code. A bin count of 64 was applied for gray-level analysis. Successively, we assessed the influence of gray-level features on the average relative percentage error by setting the bin count to 1. Finally, the analyses were repeated on a further dataset of six zebrafish to confirm the results obtained.

### 2.3. The Case Study

The case study validated the extended version of matRadiomics by assessing its ability to distinguish between two classes, high and low malformation, in images of zebrafish larvae at an early time point of 72 h post-fertilization (hpf), following radiotherapy treatment at 30 Gy [50]. Classification into high vs. low malformation was based on current scoring systems in the literature, which rely on a qualitative, operator-dependent assessment of malformations in treated larvae at 96–120 hpf [36,37,38,39]. So, the decision to use a radiomics tool in this study stems from the need to extract quantitative imaging features from optical images, which can capture complex patterns indistinguishable by the human eye alone, and which can give a more objective assessment of treatment-induced tissue changes, thus improving the accuracy and reliability of the toxicity assessment.

Preclinical studies on animal models allow us to evaluate the level of toxicity of a treatment on a complex living organism. In this context, zebrafish larvae have become a valuable model organism in toxicology studies due to their genetic similarity to humans, allowing for the extrapolation of findings to potential human outcomes. Zebrafish share approximately 70% of their genes with humans [51], making them particularly suitable for investigating the effects of drugs and therapeutic treatments. Their transparent embryos and rapid development also facilitate real-time observation of toxic effects, which manifest through visible changes in their morphological characteristics. In biomedical engineering, zebrafish are increasingly used to study drug toxicity, disease mechanisms, and the efficacy of novel treatments. The small size, ease of genetic manipulation, and high reproductive rate of zebrafish provide a practical and cost-effective system for large-scale screening of compounds [52].

Several studies have identified key morphological changes in zebrafish larvae as reliable indicators of toxicity. These visible abnormalities are critical markers for evaluating the toxic impact of compounds. The embryo’s development is extremely rapid, with all major organs and tissues, most of which have comparable position and function to human ones, fully formed by 48 hpf [53]. This consideration leads us to perform a very detailed and significant post-treatment phenotypic analysis in a very short time window. To effectively assess these changes, specific organs and structures in the zebrafish larvae have been identified as the primary focus for toxicity analyses. The organs of interest include the (i) eye, (ii) head, (iii) jaw, (iv) heart, (v) yolk, (vi) swim bladder, (vii) body length, (viii) and curvature of the spine [54]. Although each malformation is indicative of induced toxicity, as slowed or impaired development, they do not all have the same weight. Alterations such as the curvature in the spine (spinal curve, SC) or a reduction in the diameter of the head, as well as an increase in the diameter of the heart sac (Pericardial Edema, PE), are the most interesting ones, strictly correlated with the future survival of the larva, and they have a pathological counterpart in humans [52]. The spine is the most typical characteristic of vertebrates, and it is essential for survival and reproduction. The most common spinal disorder in humans is spinal curvature, and zebrafish have several advantages for modeling human scoliosis. Firstly, there is a high similarity in the morphology and structure of the spine, with a high degree of genetic conservation. Secondly, it is hypothesized that the mechanical force generated during swimming is loaded on the spine in a similar way to that in humans for posture and movement [55]. It follows that the presence of the SC in zebrafish leads to an impairment in mobility and swimming ability. Microcephalia in zebrafish is recognizable when the head is disproportionately small compared with other body structures, and it represents, as well as in humans, a disorder characterized by significantly reduced brain size and neurological development [56]. Zebrafish have proven useful for modeling human heart diseases due to the similarity of zebrafish and mammalian hearts. In addition, embryonic cardiac development is rapid, and its function is easy to observe and quantify. As with heart failure in humans, PE may lead to death in zebrafish larvae [57].

Radiomics tools, such as matRadiomics, can further enhance the precision of these toxicology studies by quantifying morphological changes and correlating them with treatment outcomes. Indeed, by adapting radiomics methodologies to preclinical models, this preliminary application opens new avenues for translational research, providing a robust platform for evaluating drug safety and understanding the mechanisms underlying toxicity, with potential implications for human health.

#### 2.3.1. The Dataset

The dataset consists of two datasets in JPEG format that are organized according to the degree of malformation of the zebrafish. According to scoring systems described in the literature and based on the analysis of the whole larva [36,37], the larvae presenting scores 1–2 were included in the low malformation group, while those presenting scores 3–4 were included in the high malformation group. Specifically, the belonging of a larva to one score rather than another, from the value of 1 to the value of 4, derives from the presence of single malformations classified as mild (1) vs. multiple mild malformations/a single moderate malformation (2) vs. multiple moderate malformations/a single severe malformation (3) vs. multiple severe malformations (4). The high-malformation dataset consists of 82 images, while the low-malformation one consists of 61 images. Following the literature on the use of zebrafish in toxicological analysis [39], six masks were analyzed for each zebrafish: whole fish, eye, heart, yolk, head, and length. The experimental protocol described in this study was carried out exclusively with zebrafish embryos and larvae up to 120 hpf. At this stage of their life cycle, zebrafish are not capable of independent feeding and, therefore, are not subject to the Italian (D.lgs 26/2014) and European (2010/63/EU) rules on the protection of animals used for scientific purposes.

#### 2.3.2. Performance Evaluation and Statistical Analysis

Classification performance was evaluated using the area under the ROC curve (AUC) and accuracy for each mask type and ML classifier. The AUC value, ranging from 1 to 0.5, reflects the quality of class discrimination, with values close to 1 indicating strong discrimination, and those near 0.5 suggesting random performance. Accuracy captures classification correctness by accounting for true positives and true negatives. To compare resampling and resizing methods in image preprocessing, a two-tailed paired-sample *t*-test was performed based on $\delta_{i}$, as defined in Section 2.2, with a *p*-value < 0.05 indicating statistical significance.

## 3. Results

### 3.1. Resolution Analysis for Radiomics Feature Extraction

To identify the optimal settings for the radiomics study, ensuring faster feature extraction without compromising the integrity of the features, two different preprocessing methods were used with a reduction factor of 2.5: (i) loading resized images and (ii) applying the matRadiomics resampling tool after loading the images in their original size.

Figure 2 shows that the resampling method gives better results than resizing. In addition, the error with the resampling method (M = 38.22%) is lower than the resizing one (M = 79.96%) for 81 out of 107 values. The two-tailed *t*-test for paired samples yielded a *p*-value of << 0.05, suggesting that the observed differences were statistically significant and not due to random variation. Consequently, resampling proved to be the most effective preprocessing method. However, the total average error remained high. The largest deviations were observed in gray-level features, prompting us to alter the bin count to one, thereby eliminating the extraction of certain gray-level-related features to see if this reduced the error. This adjustment resulted in smaller mean relative errors compared to using a bin count of 64. Figure 3 shows the mean relative error graphically, while the Table 1 shows the total average error for bin count equal to 1 and 64. The mean relative errors of the features with the bin count equal to 1 are smaller than those with bin count equal to 64 for 90 out of 107 values. The two-tailed *t*-test for paired samples yielded a *p*-value of 0.0016. Also in this case, the observed differences were statistically significant and not due to random variation. Consequently, the bin count equal to one proved to be the best parameter to reduce the variations compared to the original images. The total average error is smaller than the previous case of analysis, leading to the conclusion that the gray-level features have a great impact on features variation.

To further refine the analysis, we evaluated the variation in feature estimation on an additional six-zebrafish dataset, again using small-sized masks. The previous results were confirmed, with an average error of 16.96% in this case. Figure 4 shows that the results for the two groups are very similar, allowing us to conclude that the setting previously identified is valid for our case study. Given the lack of established guidelines on this issue, our analysis suggests that, for an initial preclinical radiomics study based on zebrafish, the optimal settings are those reported in Table 2.

In this way, we reduced computation time while keeping the extracted features as close to their original form as possible, thus enhancing workflow efficiency without sacrificing feature validity. Reducing the error ensures a high similarity to the original features, strengthening the consistency and reliability of our analysis. Table 3 shows the features where the average relative error was greater than 2%.

### 3.2. Case Study

Six regions of interest (ROIs) were analyzed for each zebrafish: the whole fish, eye, heart, yolk, head, and length, as depicted in Figure 5. The user manually segmented each area, and for the first time, a unique name was assigned to each area (see Figure 6a), simplifying identification and tracking in subsequent stages of the radiomics workflow. After the analysis of the first image, the mask names were stored, allowing for easier selection from a list in subsequent analyses, as illustrated in Figure 6b.

For the feature extraction part, the settings given in the feature variation analysis part were used (see Table 2). In this way, 107 features were obtained. At the end of the extraction of all the features, different types of analyses were carried out: a first analysis based on all the masks, and a second analysis based on each type of mask.

#### 3.2.1. Feature Selection

Feature selection allows us to reduce the number of features by identifying only the most important ones for the study, avoiding overfitting problems (see an example in Figure 7). Model interpretability is relevant to feature selection and reduction, ensuring that users understand how features contribute to outcomes [58]. By repeating the PBC method for each study, we obtained the selected features. Table 4 shows the features selected for all analyses. Features related to morphology were commonly selected, consistent with the previous literature. However, some masks revealed unexpected features as being highly relevant, such as texture features that capture patterns and relationships in pixel intensities, providing valuable insights into tissue heterogeneity. These findings suggest that a radiomics analysis can uncover novel biomarkers or patterns that go beyond traditional features, offering new insights into the underlying biological processes. This reinforces the versatility and depth of radiomics in providing data that may not be captured by conventional analysis methods. The selection process also highlighted the importance of model interpretability, ensuring that users can trace how specific features contribute to the final predictions [36,37,38,39].

#### 3.2.2. Machine Learning

All the classifiers proposed by the program were used to train the predictive model, and k-fold stratified cross-validation with k=10 was used for validation. The ML algorithms integrated into matRadiomics are the default implementations from sklearn, the open-source ML and data modeling library for Python. For the KNN classifier, a k = 10 was chosen. This choice was made to balance model accuracy with stability. The k value represents the number of nearest neighbors used to classify each data point. Setting k = 10 provides enough neighborhood information to reduce classification variance, smoothing outliers and making predictions more reliable than lower values of k that can lead to overfitting and high sensitivity to noise. Additionally, k = 10 is often recommended based on empirical tests on datasets, where moderate values generally yield a good performance without causing excessive computational complexity. An example of the performance result (receiver operating characteristic (ROC) curve, accuracy, and confusion matrix) is shown in Figure 8.

The classification results for each type of mask and classifier are shown in Table 5, including the area under the AUC and accuracy.

The classifiers, namely LDA, KNN, and SVM, exhibit varying degrees of success depending on the type of mask analyzed. For example, the whole-fish mask generally produced higher AUC scores across all classifiers, whereas individual masks (such as eye or heart) yielded more variable results. This observation suggests that comprehensive masks encompassing larger regions may be more reliable in predicting outcomes, likely due to the broader range of information they capture. Among the classifiers, SVM exhibited a slightly better performance compared to LDA and KNN, particularly for the whole-fish mask and length mask, with an AUC of up to 0.723. The performance of each classifier remains highly dependent on the specific features selected, as evidenced by the variation across masks. These results reinforce the importance of a careful feature selection to avoid overfitting. Figure 9 summarizes, for greater clarity, the best results achieved for each mask, along with the corresponding selected features and the predictive model that demonstrated the highest performance.

## 4. Discussion

In this study, we extended matRadiomics to the preclinical domain, specifically for the analysis of high-resolution images, such as microscopy images [59], 2D spheroid images [60], and zebrafish images. The case study presented here, based on zebrafish model, demonstrates the adaptability of the platform to non-clinical image formats and its capability to *early* discriminate biological outcomes in a preclinical context. matRadiomics retains its core functionality while introducing new functions that make it a viable tool for preclinical imaging studies. This is important especially because there are currently no guidelines or tools in the literature that allow for the application of radiomics to preclinical studies. About our case study, existing *scoring* systems for the zebrafish model are based on two analysis modes. The first is the one we used for the image’s classification, and it is based on the observation of the entire larva following the parameters already described in Section 2.3.1. The second performed a qualitative–quantitative evaluation of the individual malformations (and not their coexistence), classifying them according to a dimensional parameter in relation to other body districts or, for some of them, performing appropriate measurements using the ImageJ Software [36,37,38,39]. It results in the greater probability of a wrong evaluation between different operators, and it is also highly time-consuming, resulting from the measurements of the individual images. Furthermore, the literature showed that this type of evaluation is performed during the last time points of analysis, i.e., 96 and 120 hpf [36,37,38,39], which are also those in which the *larvae* are more mobile and active [40]. The *idea*, therefore, of carrying out an automated analysis on images acquired at 72 hpf could lead both to concrete, reliable, and replicable data and to the possibility of predicting, at an early and easier time point for image acquisitions, how toxic a treatment may be, without necessarily reaching the final time points. For this purpose, we distinguish between high and low malformations in zebrafish, as reported in Section 2.3: high malformations refer to cases with more severe or numerous developmental abnormalities in zebrafish, while low malformations indicate fewer or less observable defects. This distinction helps to categorize the severity of developmental problems, allowing us to analyze the predictive power of the proposed radiomics workflow.

Our analyses started by assigning unit values for voxel spacing and slice thickness, since the actual information was not available but needed to ensure the feature extraction process. These parameters influence the value of several features related to the shape, volume, and texture. In the case where this information is lacking, we believe that a unitary setting is the best choice in terms of features interpretation. However, we acknowledge that this assumption may introduce bias, as it may not accurately represent the actual geometry of the imaging data. This simplification may affect the quantitative analysis and subsequent predictive modeling. Future studies may benefit from acquiring accurate pixel information to improve the validity of the results.

Successively, we identified the optimal settings for the radiomics study, as reported in Table 2, ensuring a faster extraction of the features without compromising their integrity. Figure 2 suggests that the resampling method significantly improves computational efficiency during feature extraction compared to resizing (average relative percentage error of 38.22% and 79.96% for the resampling and resizing methods, respectively). This highlights the need for efficient preprocessing in high-resolution preclinical images, where computational demands can pose challenges when dealing with large amounts of data. Figure 3 shows how using different bin counts for gray-level analysis also had a considerable impact on the extracted features, with smaller bin counts (i.e., one) producing more reliable and consistent results. This finding indicates that adjusting bin count could be a potential strategy for reducing error when analyzing preclinical images. However, despite improvements in error reduction, further studies are required to assess how these methods impact other feature types beyond gray-level features, such as texture-based or high-order radiomics features. The high average errors observed, particularly with the resizing method (79.96%), raise concerns about the reliability of predictions when using these preprocessing techniques. These errors may affect the consistency of feature extraction, potentially undermining the accuracy of the final model. While the resampling method shows a better performance, the substantial error differences highlight the need for further optimization. To mitigate these issues, future studies should focus on refining preprocessing techniques, such as experimenting with different resampling parameters or exploring alternative methods that may offer better precision without sacrificing efficiency. To the best of our knowledge, there are currently no reference data in the literature regarding the impact of resampling and bin count variation on the computational efficiency during feature extraction. This work is the first to investigate these aspects and to propose an effective strategy.

The final goal of our case study analysis was to evaluate the ability of matRadiomics to distinguish between high and low malformations *early* in zebrafish models. Morphological features were predominantly selected, aligning with prior studies [36,37,38,39]. However, some masks highlighted unexpected features, such as texture-based features, which capture patterns in pixel intensities and provide valuable insights into tissue heterogeneity, as shown in Table 4. This indicates that radiomics can reveal novel biomarkers or patterns beyond traditional features, offering a deeper understanding of biological processes. Table 5 shows the potential of matRadiomics as an early decision support tool in preclinical research (AUC of 0.723, whole-fish mask-based SVM model). The best result based on the SVM model analyzing the whole-fish mask could be explained by the scoring method used to divide the images into two classes (high malformation vs. low malformation), as it evaluates the entire larva [36]. This comprehensive evaluation considers the potential presence of multiple malformations across the fish. As such, it makes sense that the best-performing model is the one that examines the entire fish image, as it aligns more closely with the scoring criteria that reflect the overall condition of the larva rather than the entity of the individual alterations. Nevertheless, considering how each fish district could have a different weight for the future survival of the individual post-treatment, it could be useful to carry out a second analysis, in which the high vs. low classification is carried out for each malformation separately [37].

Our study has some limitations. While the plugin integrates three ML models, it currently lacks more advanced models, such as random forests or deep learning approaches, that could improve its suitability for complex analyses. Expanding the model options would greatly improve the versatility of the tool and the accuracy assessment [61]. Additionally, the feature extraction process for high-resolution images remains to be explored; further optimization is needed to increase efficiency in handling high-resolution preclinical data. Although the zebrafish case study demonstrates the functionality of the plugin, its generalizability to other preclinical models remains untested, such as microscopy [59] and 2D spheroid images [60], to further validate its versatility. However, caution is needed when translating our findings. Variations in imaging modalities, biological markers, or species-specific differences could affect the generalizability of the methodology. For instance, different models may require adjustments in preprocessing steps or feature extraction methods. Future research should aim to validate these techniques across a wider range of species and experimental settings to establish their robustness and adaptability. Finally, the case study does not include external validation on independent datasets, an essential step to ensure the tool’s robustness and applicability across different scenarios.

## 5. Conclusions

In this study, we extended matRadiomics to the preclinical domain. Our case study on zebrafish models demonstrated the platform’s ability to discriminate biological outcomes early, a critical step in preclinical research. Our results suggest that an early automated analysis at 72 hpf can predict treatment toxicity, potentially reducing the need for more laborious subsequent assessments. Although our study highlights the potential of matRadiomics, some limitations remain. The impact of parameter settings for the feature extraction process for high-resolution images requires further investigation. Furthermore, expanding the range of ML models and validating the platform on different preclinical models are essential for its wider adoption. Despite these limitations, our work represents a significant step towards integrating radiomics into preclinical research, accelerating drug discovery and development.

## Figures and Tables

**Figure 1 jimaging-10-00290-f001:**
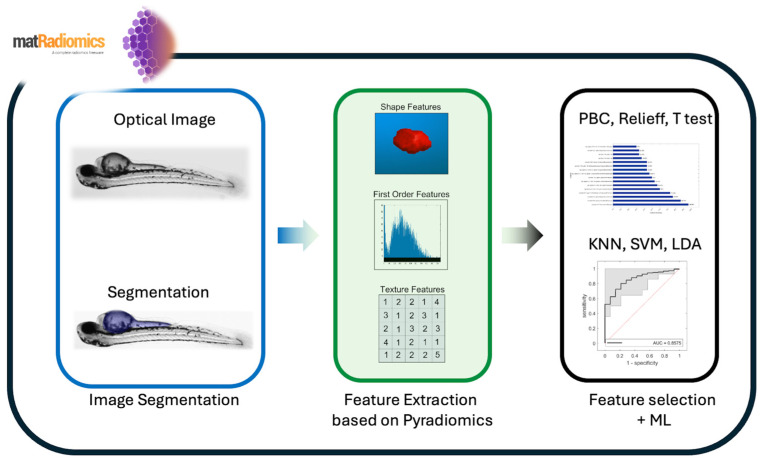
The workflow of the extended version of matRadiomics for preclinical studies.

**Figure 2 jimaging-10-00290-f002:**
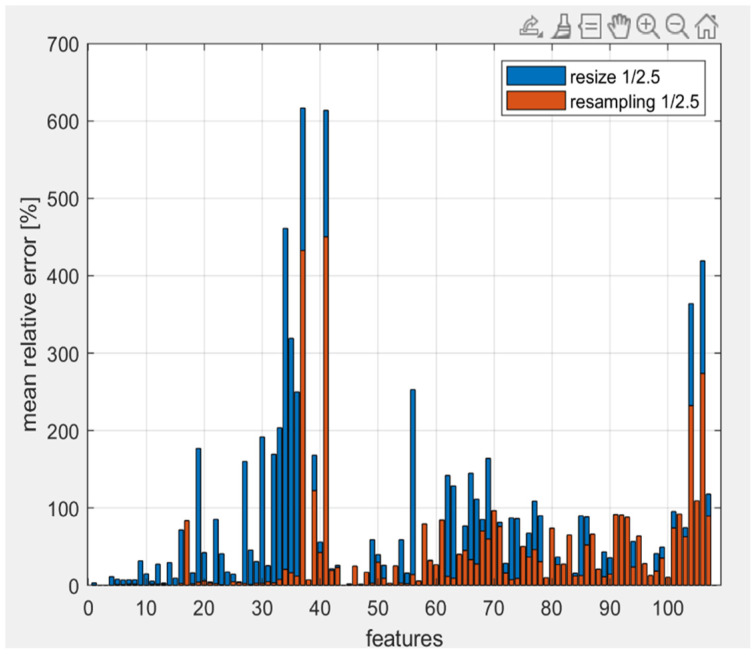
Mean relative error for each feature for the two different preprocessing methods.

**Figure 3 jimaging-10-00290-f003:**
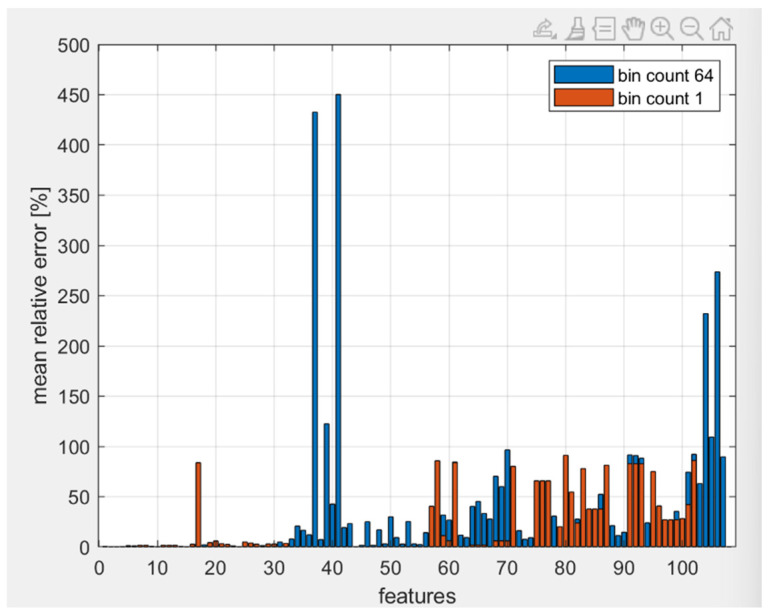
Mean relative error for bin count equal to one and equal to sixty-four.

**Figure 4 jimaging-10-00290-f004:**
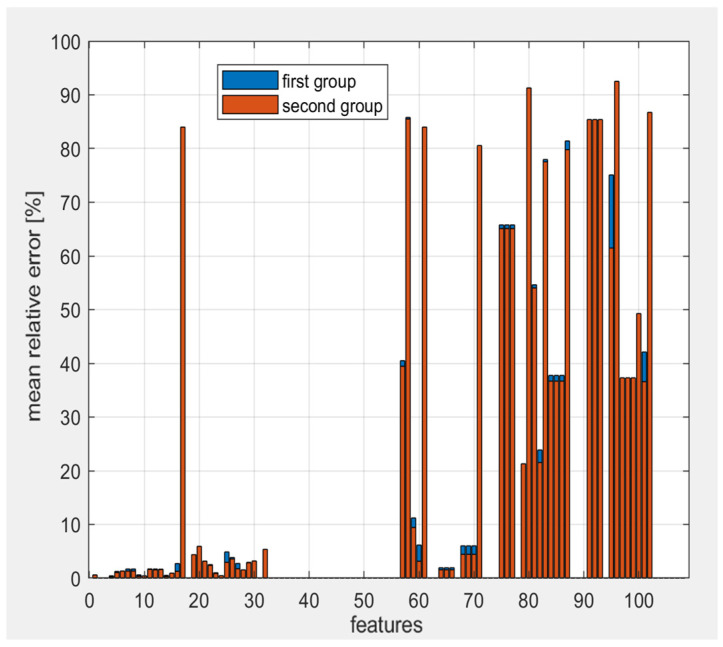
Mean relative error on features for the first and the second zebrafish dataset.

**Figure 5 jimaging-10-00290-f005:**
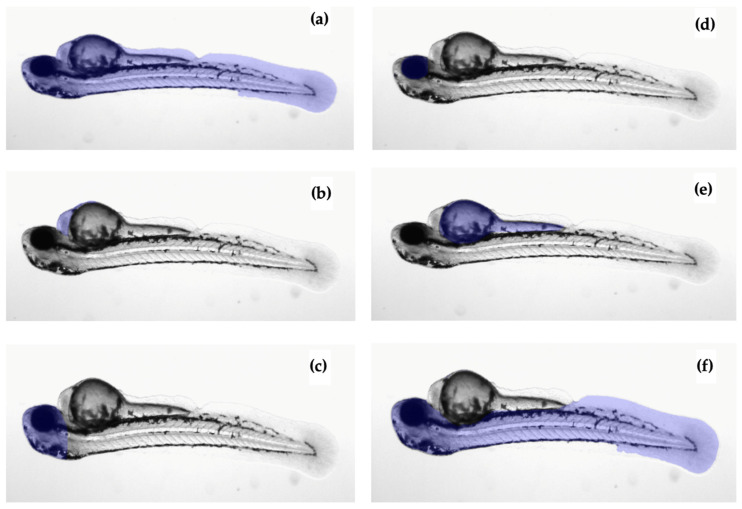
Example of all_fish (**a**), heart (**b**), head (**c**), eye (**d**), yolk (**e**), and length (**f**) masks used in the study.

**Figure 6 jimaging-10-00290-f006:**
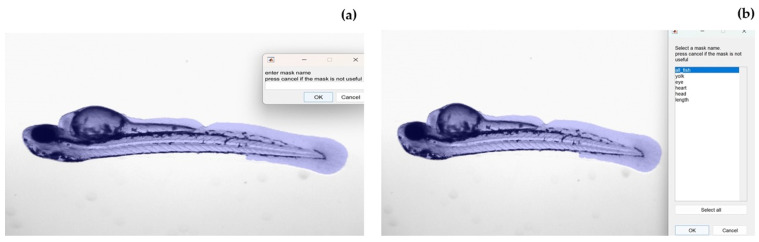
(**a**) The window that appears to manually assign the name of the mask for the analysis of the first image. (**b**) The window with the list of masks used after the extraction of the features of the first image.

**Figure 7 jimaging-10-00290-f007:**
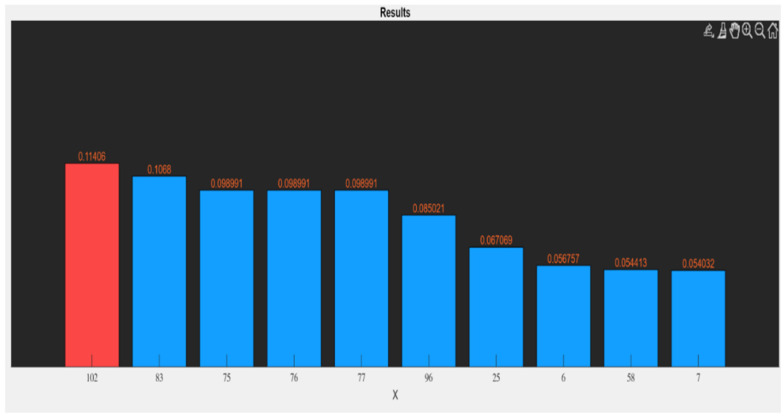
Example of bar plot for selected features for all masks without all_fish mask. In red, the feature selected using the PBC method.

**Figure 8 jimaging-10-00290-f008:**
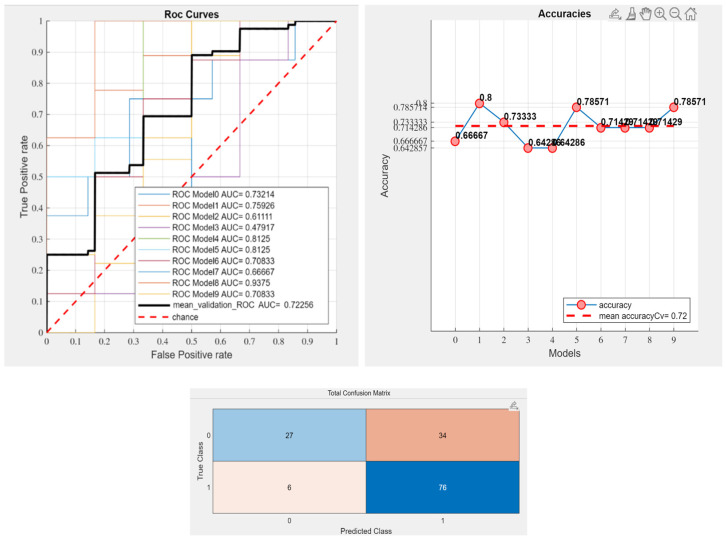
Example of performance of the predictive model for the all_fish mask based on the ROC curve, precision and confusion matrix.

**Figure 9 jimaging-10-00290-f009:**
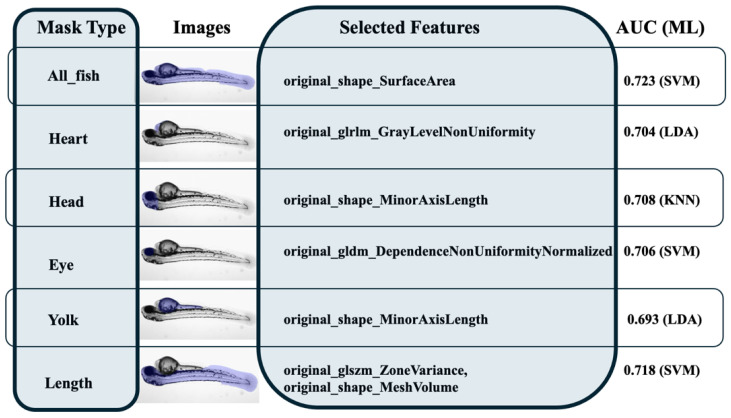
The best results obtained for each mask, together with the corresponding image, the selected features, and the predictive ML model that achieves the highest performance.

**Table 1 jimaging-10-00290-t001:** Parameters for feature variation in the case of gray-level analysis.

Resampling	Bin Count = 1	Bin Count = 64
Total average error	16.04%	38.22%

**Table 2 jimaging-10-00290-t002:** Parameter setting used in our preclinical study.

Preprocessing	Resampling Technique	Bin Count
Resampling, 2.5 factor	Stick-nearest neighbor	1

**Table 3 jimaging-10-00290-t003:** Features with an average relative error major than 2%.

Features with Error > 2%
“original_firstorder_Energy”
“original_firstorder_Kurtosis”
“original_firstorder_Skewness”
“original_firstorder_TotalEnergy”
“original_gldm_DependenceNonUniformity”
“original_gldm_DependenceVariance”
“original_gldm_GrayLevelNonUniformity”
“original_glrlm_GrayLevelNonUniformity”
“original_glrlm_LongRunEmphasis”
“original_glrlm_LongRunHighGrayLevelEmphasis”
“original_glrlm_LongRunLowGrayLevelEmphasis”
“original_glrlm_RunEntropy”
“original_glrlm_RunLengthNonUniformity”
“original_glrlm_RunVariance”
“original_glszm_GrayLevelNonUniformity”
“original_glszm_LargeAreaEmphasis”
“original_glszm_LargeAreaHighGrayLevelEmphasis” “original_glszm_LargeAreaLowGrayLevelEmphasis” “original_glszm_SizeZoneNonUniformity”
“original_glszm_ZoneEntropy”
“original_glszm_ZoneVariance”

**Table 4 jimaging-10-00290-t004:** Selected features for each type of mask.

Mask Type	Selected FEATURES
All masks	original_glszm_ZoneVariance
All_fish	original_shape_SurfaceArea.
Eye	original_gldm_DependenceNonUniformityNormalized
Head	original_shape_MinorAxisLength
Heart	original_glrlm_GrayLevelNonUniformity
Length	original_glszm_ZoneVariance, original_shape_MeshVolume
Yolk	original_shape_MinorAxisLength

**Table 5 jimaging-10-00290-t005:** Performance of predictive model for all the analyses. The best result obtained is in bold.

Mask Type	LDA	KNN	SVM
All masks except all_fish	AUC: 0.568 Acc: 0.582	AUC: 0.576 Acc: 0.558	AUC: 0.464 Acc: 0.591
All_fish	AUC: 0.720 Acc: 0.679	AUC: 0.684 Acc: 0.628	**AUC: 0.723** Acc: 0.72
Eye	AUC: 0.702 Acc: 0.658	AUC: 0.686 Acc: 0.608	AUC: 0.706 Acc: 0.629
Head	AUC: 0.692 Acc: 0.663	AUC: 0.708 Acc: 0.651	AUC: 0.683 Acc: 0.650
Heart	AUC: 0.704 Acc: 0.670	AUC: 0.696 Acc: 0.671	AUC: 0.677 Acc: 0.670
Length	AUC: 0.712 Acc: 0.692	AUC: 0.670 Acc: 0.638	AUC: 0.718 Acc: 0.693
Yolk	AUC: 0.693 Acc: 0.665	AUC: 0.613 Acc: 0.622	AUC: 0.623 Acc: 0.629

## Data Availability

Source code, documentation, and examples are available upon free request to the authors. After the publication of the article, the software will be available on the institutional websites of the authors (IBSBC-CNR and Sapienza University).

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
