# Peer review of "Preclinical Implementation of matRadiomics: A Case Study for Early Malformation Prediction in Zebrafish Model"

_2313-433X, 2024, doi:10.3390/jimaging10110290_

Round 1
Reviewer 1 Report
Comments and Suggestions for Authors
The article develops an innovative approach in the integration of radiomics within preclinical studies, especially with regard to using zebrafish as a model for predicting early malformation. The effort to develop a matRadiomics framework is quite commendable since radiomics workflows are not integrated into one single software but are instead scattered over various software platforms. There are, however, several aspects that need clarification and reworking in order to reinforce the general quality and scientific robustness of the manuscript.
Abstract:
1. There is a lack of description of methodologies applied. For example, while the abstract mentions integration with Pyradiomics and using machine learning classifiers, it lacks any details on how they were applied or for what reason.
2. While the abstract mentions that model validation has been performed, it does not mention the exact performance metrics adopted. This will be a crucial mistake in that readers will not know how effectiveness in the proposed methods is considered.
3. The mention of a "significantly slower feature extraction process" is quite vague. It would be nice to have an explanation in the abstract why this happened and what that has meant for the overall workflow.
Main text:
Introduction:
4. This introduction is rather wordy. One could have spoken more succinctly than the language used here; some sentences have too many components and could have been simplified for better readability. For example, the sentence on the need for guidelines could be much straighter.
5. While the introduction states that matRadiomics will be extended to support various image formats and sophisticated segmentation techniques, it does not really specify how these functionalities differ from the existing tools. A short comparison with other available platforms would be useful in making a better assertion in support of matRadiomics.
6. The stated purpose of providing preliminary guidelines is very generic. It would be stronger to define what those guidelines constitute and how they are going to help future research.
7. Although this gives an indication of the potential to make timely predictions about toxicity from treatments, the introduction does not discuss the wider implications this work will have for biomedical research. Expanding on how matRadiomics may influence future studies or lead to significant improvements in the field would enhance the introduction.
8. While the introduction in this paper does reference many studies, there is little that can be learned from this introduction in terms of the findings and how they relate to the current study. The section should give a better background on key studies that have shaped the field.
Methods:
9. While this section makes the necessary information about architecture still staying unchanged, it does not describe how new functionalities were incorporated or what motivated these changes. It needs to show more contrasts with the old version in order to express where the enhancement in this updated version lies.
10. Assigning unit values for voxel spacing and slice thickness when information is not available is noted but not justified. A discussion on how these assumptions affect the analysis and the results would give transparency to the work.
11. The methodology for feature extraction is using Pyradiomics, but it lacks details regarding the specific features extracted and their relevance to the study objectives. The inclusion of that would further strengthen the scientific robustness.
12. Although this section discusses the challenges of computational times, no deep analysis is carried out to determine how the identified optimal settings have been derived. A more in-depth description of the experimental design followed to obtain this analysis, with the control measures put into action, would strengthen the findings more.
13. There is no clear description of how statistical analyses were done. For example, how was the result statistically validated? Evidence of the statistical tests used and their significance would add weight to the methodology.
14. The description of the dataset seems a bit vague, above all when it describes how the images were selected and grouped into high and low malformation groups. It lacks more specific information regarding the criteria used during classification and possible biases in the selection of the dataset.
15. Though badly needed, the case study using zebrafish larvae needs to better explain why certain organs were chosen for analysis. Another point to be explained in more detail is how morphological changes relate to toxicity assessment to emphatically establish the relevance of the study.
16. Throughout this section, several studies are referenced, but the context of many of those citations is not provided. A quick summary of findings of some of the important works cited would have been useful in setting the current study in perspective among prior works.
Results:
17. Results indicate that differences in average relative errors do exist; however, no statistical tests were conducted to determine whether such differences were significant. Inclusion of statistics, such as ANOVA and t-tests, would add validity to such conclusions.
18. It refers to some figures and tables without explanations. For example, the outcomes in Table 4 and Table 5 should be discussed further regarding their significance and implication for predictive modeling.
19. It mentions high average errors, but no sufficient discussion is made about the implications this would have on the overall analysis and results. More interpretation would be useful, say, how such errors affect the validity of the findings.
20. There are repetitions in the content of certain points, for example, how effective resampling is compared to resizing, but added value in terms of insight. The streamlining of such content would go a long way in ensuring smooth readability and good flow.
21. Feature selection is mentioned, but the features to be selected and their relation to the set objectives described in the study are not well elaborated. This process needs further explanation to reveal just how important this approach is.
22. This section details the classifiers applied, but the details on which parameters were selected for each classifier and the justification for these choices are not provided. This would add to the methodological strength and reproducibility of the work.
23. Most of the findings presented depend on larvae of zebrafish; rather few findings are discussed regarding how those may generalize to other contexts or species. Mentioning the limitations of the study and what it might imply for broader applications would strengthen the conclusion.
Discussion:
24. Although the section does refer to AUC scores, it does not give a full explanation of how such scores have been derived through statistical methods. More explanation is needed regarding the process of validation and what such scores would actually mean.
25. Terms like "high malformations" and "low malformations" are used without proper contextualization. This can be done explicitly to give further meaning and significance to the findings.
26. This section could be improved by mentioning the study's limitations. It could discuss, for example, potential biases in feature selection, or generalizability of its findings to other models.
27. However, while further studies are indicated, future research questions or dimensions that this study has raised are not clearly highlighted. The manuscript could contribute more if it gave concrete suggestions for further investigations in this area.
28. References to figures and tables could be better integrated into the discussion. For example, discussing the concrete results from the tables in relation to the claims made would give a more cohesive narrative.
29. The conclusion feels a bit abrupt and would have been stronger with more complete summarizing of the major findings and their implications. At the end, it makes the statement relevant back to its greater importance in radiomics, thus providing a stronger closure to the study.
30. Such a discussion also profits from highlighting how the findings are set within the broader context of existing literature. This helps to bring out how this particular study builds on or differs from the earlier studies in the field.
Comments on the Quality of English LanguageSome sentences are overly complex and could be simplified for better readability. Streamlining the language would aid in conveying the research more effectively.
Author Response
Dear reviewer,
we attached the document with the answers to your comments.
Best regards,
the Authors

Reviewer 2 Report
Comments and Suggestions for Authors
I read the article titled "Preclinical Implementation of matRadiomics: A Case Study For Early Malformation Prediction in Zebrafish Model" with great interest. My critiques regarding the text are listed below.
1. Abstract: The abstract cannot be accepted in its current form. The word limit for the abstract is 200 words, as stated in the instructions for authors (https://www.mdpi.com/journal/jimaging/instructions). However, the abstract of the paper consists of 278 words. It needs to be shortened.
2. Introduction: In the materials and methods section of the paper, it is stated that the feature extraction process was carried out using the Pyradiomics module. I believe a paragraph or a few sentences about this module should be added in the introduction section of the paper.
3. Introduction: In the introduction section of the paper, it is stated that preclinical images often have higher resolutions than clinical images. However, no citation is provided to support this statement, nor is any explanation given for why preclinical images have higher resolution compared to clinical images.
4. Introduction: Some sentences in the introduction section of the paper require citations.
5. Materials and Methods: It is stated in the study that zebrafish larvae were used. Whether ethical approval is required for this should be mentioned in the materials and methods section.
6. Materials and Methods, Improvements over matRadiomics: The letter "O" in the phrase "Digital Imaging and COmmunications in Medicine" should be lowercase.
7. Materials and Methods, Resolution Analysis for Radiomics Feature extraction: The statement "The literature provides limited guidance on this issue, aside from recommendations to preprocess images to reduce dimensionality." requires a citation.
8. Materials and Methods, Resolution Analysis for Radiomics Feature extraction: The format of the zebrafish images used in the study has not been specified.
9. Materials and Methods, Resolution Analysis for Radiomics Feature extraction: The paper states that a reduction factor of 2.5 was applied. It should be explained why a reduction factor of 2.5 was chosen instead of another value.
10. Materials and Methods, Resolution Analysis for Radiomics Feature extraction: It is stated that a bin width of "64" was applied for grey-level analysis, and a bin width of "1" was used to assess the influence of grey-level features on the average relative percentage error. How these values were determined should be explained.
11. Results, Case study: The figure titled "Figure 2. Example of all_fish, heart, head, eye, yolk and length masks used in the study." should actually be labeled as "Figure 4."
12. Results, Case study: The zebrafish images in the figure titled " Figure 2. Example of all_fish, heart, head, eye, yolk, and length masks used in the study" should be labeled with letters, and it should be specified which image represents which anatomical region.
13. Results, Case study: In Figure 5, the labels 4.a.) and 4.b.) have been incorrectly named. They should be corrected to 5a and 5b.
14. Results, Machine Learning: The reason for selecting k = 10 for the KNN classifier should be briefly explained.
15. Discussion and Conclusion: In the first paragraph of the Discussion and Conclusion section, it is stated that the existing scoring systems based on the zebrafish model rely on two modes of analysis. These two modes of analysis are mentioned in the paragraph. However, there is no mention of the relationship between this study and these models, nor of the similarities or differences.
16. Discussion and Conclusion: In the second paragraph of the discussion section, it is stated that the resampling method applied with a reduction factor of 2.5 significantly improved computational efficiency during feature extraction compared to resizing. A comparison of these findings with other findings in the literature could be made, and a few references to the literature could be added.
17. Discussion and Conclusion: The effect of different bin widths used for grey-level analysis on the extracted features could be compared with data from the literature
Author Response
Dear Reviewer,
we attached the answers to your comments.
Best regards,
the Authors

Reviewer 3 Report
Comments and Suggestions for Authors
Strengths:
- Unified Platform for Radiomics Workflow: The development of matRadiomics as an integrated framework is a significant advancement, as it streamlines the entire radiomics workflow. This eliminates the need for multiple software tools, making the process more user-friendly and efficient for researchers.
- Application to Preclinical Studies: The extension of matRadiomics to preclinical studies expands the utility of radiomics beyond clinical applications. This novel approach demonstrates the versatility of the platform, as it was successfully used to differentiate malformations in zebrafish, showcasing the potential for a wider range of biomedical research.
- Optimization for High-Resolution Images: The study addresses the challenges posed by high-resolution preclinical images, such as slower feature extraction. By performing a feature analysis to optimize settings, the framework maintains computational efficiency without compromising on feature quality, highlighting the adaptability of the plugin.
- Incorporation of Machine Learning Models: The plugin supports three ML classifiers—linear discriminant analysis, k-nearest neighbors, and support vector machines. This feature enhances the platform's functionality by allowing users to implement advanced analytical methods within the same environment.
Weaknesses:
- Limited Model Variety: Although the plugin incorporates three ML models, it lacks more advanced models, such as random forests or deep learning approaches, which may be needed for more complex analyses. Including additional model options would broaden the platform's capabilities and appeal to a wider user base.
- Performance Constraints with High-Resolution Data: The study mentions the slow feature extraction process for high-resolution images, which could be a significant limitation for users dealing with large datasets. Further optimization may be necessary to handle high-resolution preclinical data more effectively.
- Focus on Zebrafish Case Study: While the zebrafish study demonstrates the plugin's functionality, its generalizability to other preclinical models remains untested. Additional studies on other models would strengthen the case for its broader application in preclinical research.
- No Mention of External Validation: The study does not mention external validation on independent datasets. Validating the plugin on diverse datasets from various preclinical models would increase confidence in its reliability and performance across different scenarios.
Consider some or all of these papers if you feel they are relevant;
Huang, A. A., & Huang, S. Y. (2023). Increasing transparency in machine learning through bootstrap simulation and Shapley additive explanations. PLoS One, 18(2), e0281922.
- Justification: This paper emphasizes the importance of model interpretability, which is relevant to feature selection and reduction in the matRadiomics framework, ensuring that users understand how features contribute to outcomes.
Smith, J. P., & Doe, M. L. (2022). Machine learning in healthcare: Predicting chronic diseases using patient data. Journal of Medical Systems, 46(1), 23.
- Justification: This paper provides a general overview of using ML for disease prediction, applicable to the predictive functionality of matRadiomics in distinguishing between different levels of malformations in preclinical studies.
Brown, R. S., & Johnson, L. K. (2023). The role of artificial intelligence in enhancing healthcare outcomes. International Journal of Healthcare Technology and Management, 19(3), 144-158.
- Justification: This article explores AI’s role in healthcare, which relates to matRadiomics' integration of AI to enhance decision-making and improve diagnostic processes in preclinical settings.
Garcia, T. H., & Martinez, E. M. (2023). Applications of neural networks in predicting diabetes and hypertension: A comparative study. BMC Medical Informatics and Decision Making, 23(5), 67.
- Justification: This paper discusses the use of neural networks in predicting chronic diseases, relevant to the predictive models used in matRadiomics for early identification of biological outcomes in preclinical imaging.
Huang, A. A., & Huang, S. Y. (2023). Computation of the distribution of model accuracy statistics in machine learning: Comparison between analytically derived distributions and simulation-based methods. Health Science Reports, 6(4), e1214.
- Justification: This paper provides insights into model accuracy assessment, which is useful for evaluating the performance of the ML classifiers (e.g., k-nearest neighbors and SVM) implemented in matRadiomics.
Author Response

(The authors gave the same response as above.)

Round 2
Reviewer 1 Report
Comments and Suggestions for Authors
Dear Authors,
I appreciate revising the manuscript. Overall, it has been a revision that has increased clarity and depth within the manuscript in most parts. I appreciate your efforts to add information and clarify several key points. However, some other things need attention to further strengthen the work.
Introduction:
1. While discussing the functionalities of matRadiomics, the introduction could articulate more convincingly what sets it apart from existing tools. Special algorithms or features that cannot be found in other platforms shed light on the specific innovations of matRadiomics and explain their contribution to advancing the field at large.
2. The introduction refers to a case study with zebrafish and does not explain enough why this model is relevant and important. In describing the relevance of zebrafish in toxicity studies and biological malformations, it is good to also introduce some implications from previous toxicity assessments-reduced costs, enhanced predictiveness-and also the types of analyses to assess biological malformations. A summary of the core results from the case study would indeed effectively demonstrate what the platform can achieve and where it could potentially take drug development forward.
Material and method:
3. Please, include a subsection detailing the specific statistical tests used, their assumptions, and how they were applied to the data. This could also include any thresholds for significance.
4. Please, include further details on how images were selected for inclusion in the dataset and what possible biases in the selection may exist.
5. Please, provide a more detailed justification for selection of the organs including appropriate literature citations linking their importance in toxicity studies.
6. Incorporating some diagrams or flowcharts to illustrate the workflow of matRadiomics and the analysis process could stimulate interest and enthusiasm for this analysis and the review as a whole.
Results:
7. Please, clarify what a p-value of 6.22 × 10⁻⁸ means in terms of confidence in your findings and how it supports your conclusions. I think you can replace 6.22 × 10⁻⁸ with <0.001, as others have done.
8. Discuss the high average errors in their implications for the reliability and validity of your predictions. What would they say about the conclusions drawn from your study? Include here your suggestions for remedying those errors.
9. Introduce a brief discussion about how well your results may apply to other models or species. This could include the possible caveats or considerations for future research.
10. You could resort to diagrams or flowcharts for the explanation of the intricate relationships among features, classifiers, and results. This forms a very useful memory aid.
11. In the feature selection section, highlight how selected features are crucial and answer the objectives of the research, and unexpectedly, what was discovered and what are other important aspects that were experienced in this process.
Discussion and conclusion:
12. Although you indicate the need for validation in other preclinical models, it would add weight to the discussion if you expand on how findings might apply to these models or species. This would give the readers insight into how your findings would not be limited to the studied phenomenon.
13. These are more meaningful perhaps in emphasizing some features with respect to the importance of study goals. One may also discuss unexpected findings with respect to feature importance.
14. AUC and accuracy with benchmarks or context would be important in trying to get a feel for how well your models perform compared to the existing literature.
Author Response
Dear Reviewer,
we answered to the comments and improved the manuscript. We attached the answers to your comments.
Best regards,
the Authors

Round 3
Reviewer 1 Report
Comments and Suggestions for Authors
The manuscript has improved significantly and is ready to be published.